# Mental health among sexual and gender minorities: A Finnish population-based study of anxiety and depression discrepancies between individuals of diverse sexual orientations and gender minorities and the majority population

**Marianne Källström** [ID]*, **Nicole Nousiainen, Patrick Jern, Sabina Nickull** [ID], **Annika Gunst**

Department of Psychology, Faculty of Arts, Psychology and Theology, Åbo Akademi University, Turku, Finland

* mkallstr@abo.fi

**Data Availability Statement:** The data used in this study cannot be shared publicly because of the

## Abstract

Substantial empirical evidence suggests that individuals who belong to sexual and gender minorities experience more anxiety and depression than heterosexual and cisgender people. Many previous studies have not, however, used population-based samples. There is also a shortage of evidence about certain sexual and gender minorities (e.g., nonbinary individuals). We examined differences in levels of anxiety and depression within sexual and gender minorities, as well as compared to the heterosexual and cisgender majority in a population-based Finnish sample ($N = 8,589$). We also explored if individuals who belong to both a gender and a sexual minority (double minority) reported higher rates of anxiety and depression than individuals who hold either a gender or a sexual minority status (single minority). Individuals who belonged to either a sexual or a gender minority overall experienced significantly higher rates of anxiety and depression than cisgender and heterosexual individuals. Among the different sexual and gender minorities, bisexual, emerging identity, and nonbinary individuals reported the highest rates of anxiety and depression. We found no differences in anxiety and depression between single minority and double minority individuals. Our results suggest that even though Finland is a country with an inclusive social climate, sexual and gender minorities are, nevertheless, disproportionately affected by mental health issues. The present study gives further support to the claim that individuals holding a sexual or gender minority status experience higher levels of depression and anxiety compared to cisgender and heterosexual individuals and pinpoints the need to acknowledge these issues both in the context of health care and in the society at large.

sensitivity of the research topic. The data includes information on sexual orientation, gender identity, the number of siblings each participant has, and twin status (twin/singleton). Each family is assigned its own code in the data (i.e., one that is shared between all members of the same family), it means that it could be theoretically possible to identify individual participants in some cases. Finland is a small country (with a population of ca. 5.3M people), so having access to data revealing a person's age, sexual orientation, gender identity, and twin status in combination with knowledge about how many twin and non-twin siblings they have and what the siblings' ages, gender identities, and sexual orientations are can already compromise the anonymity of certain participants and families. Thus, we have opted not to share data publicly but data can be made available on request to researchers who meet the criteria for access to confidential data as well as editorial staff at journals. To request access to the data, please contact either the corresponding author Marianne Källström (mkallstr@abo.fi or +358 469217716) or the Board for Research Ethics at Åbo Akademi University (secretary Nina Bäckman, nina. backman@abo.fi) For other questions and non-author affiliated data secrecy and data handling matters, please contact the Åbo Akademi University Data Protection Officer Anna-Maria Nordman (anna-maria.nordman@abo.fi; https:// www.abo.fi/en/processing-of-personal-data-at-abo-akademi-university/)

**Funding:** This research was funded by Grants No. 319403, 284385, and 274521 from the Academy of Finland (https://www.aka.fi/en/; all awarded to P. J.). The funders had no role in study design, data collection and analysis, decision to publish, or preparation of the manuscript.

**Competing interests:** The authors have declared that no competing interests exist.

# Introduction

Depression and anxiety are two of the most prevalent types of mental disorders, and they are considered to be among the leading factors contributing to the global overall burden of disease [1]. Some societal groups, such as sexual and gender minorities, are disproportionately affected by mental health problems. In comparison to the heterosexual and cisgender majority, these minorities report higher rates of depression, anxiety, and other mental health issues [2–6].

The *minority stress theory* is the most frequently applied explanation for the discrepancy between mental health in gender and sexual minorities and mental health in the majority population [7–9]. Proponents of the minority stress theory suggest that sexual and gender minorities more frequently suffer from mental health problems due to chronic stress experienced as a result of a marginalized social status as well as accompanied institutionalized prejudice and stigma. The theory was originally developed to apply to the lesbian, gay, and bisexual population [9], and has later been adapted to include transgender populations [7]. The minority stress theory has received empirical support from multiple studies [10–14].

Recent research has aimed to extend the scope of the minority stress theory to encompass the experiences of people who hold nonbinary gender identities, hypothesizing that nonbinary individuals face the same structural stressors as binary transgender individuals but in more intense and harmful ways [15–18]. This argumentation rests partly on the fact that most societies are built on a two-gender system that makes many situations harder to navigate if one does not identify as either male or female (e.g., gendered pronouns and bathrooms). Because of the centrality of this two-gender system, understanding what it is to be nonbinary is likely harder for the general public than understanding the experience of a transgender person who wishes to affirm the distinctly male or female gender they belong to regardless of the body they have been born with. As nonbinary gender identities have not been represented in media or research until recently, nonbinary individuals are also likely to face more ignorance and ignorance-related prejudice than individuals who hold a binary transgender identity. In order to empirically test whether being nonbinary is associated with more minority stress, Lefevor and colleagues [15] analyzed a large student sample and detected that although transgender and genderqueer individuals both experienced more stress and anxiety than cisgender individuals, the group of genderqueer participants displayed more stress and more anxiety compared to the binary transgender group. Thorne and colleagues [18] as well as Scandurra and colleagues [17] obtained similar results, although partly discrepant or unclear findings have also been reported [16]. Consequently, it seems probable that nonbinary individuals experience more minority stress and have more severe minority stress-related health outcomes than binary transgender individuals, even though more quantitative large-scale research is still needed.

## Prevalence and definitions

The magnitude of the different gender and sexual minorities in the overall population is difficult to determine, and available statistics vary [19]. In a review of nine population-based surveys, Gates [20] estimated that 3.5% of adults in the US identify as gay, lesbian, or bisexual. Rahman and colleagues [21] analyzed a sample of 191,088 individuals across 28 countries and concluded that 5.1% of men and 7.2% of women identified as bisexual, while 4.9% of men identified as gay and 2.1% of women identified as lesbian. Alongside the sexual orientations mentioned above, several other sexual orientation categories have gained attention and come into broader awareness during recent decades. These sexual orientations include categories such as pansexual, demisexual, queer, questioning, and asexual. Even though these categories have many differences between them and even though the meaning of several of them can also be understood in slightly varying ways, we have chosen to collectively use the term *emerging*

*identities* (EI) to refer to them in the present study. The term EI has been used in previous research concerning mental health and minority stress among gender and sexual minorities [see e.g., 22, 23], and refers to the relatively new status of these terms in the realm of mainstream awareness. Reliable and population-representative estimates of the proportion of individuals who identify as EI are limited, and many studies have used convenience samples with varying measurement methods [24, 25]. A couple of recent population-representative estimates have emerged. For example, Greaves and colleagues [26] studied a nationally representative sample from New Zealand and found that 0.5% identified as pansexual, 0.3% identified as asexual, and less than 0.1% identified as queer. Lindström and colleagues [27] found that 1.1% of the men and 0.9% of the women in a population-based study in Sweden reported that their sexual orientation was something other than heterosexual, gay, lesbian, or bisexual. There are no population-based prevalence estimates on how common the above-mentioned various sexual orientations are among Finnish adults, but a population-based research report from a recent adolescent health survey indicated that 88% identified as heterosexual, 6% as bisexual or pansexual, 1% as gay or lesbian and 5% as something else [28].

The term transgender is most frequently used as an umbrella term [29], encompassing both binary transgender individuals as well as those outside the gender binary (e.g., nonbinary or agender). Approximately 0.3% of the population in the US identify as transgender [21]. There is a shortage of reliable and population-representative estimates of how common it is to identify as nonbinary or agender. Some recent studies have, however, provided preliminary population-representative estimates. For example, Kaltiala-Heino and Lindberg [30] estimated that 3.3% of the adolescents in a population-based Finnish school survey identified either as "both male and female", or as "neither male nor female", which may reflect nonbinary and/or agender identity. In their sample, 0.6% of the adolescent participants reported having a binary transgender identity. In a population-based study in Flanders, Belgium, Van Caenegem and colleagues [31] found that 0.7% of men and 0.6% of women experienced gender incongruence and 2.2% of men and 1.9% of women experienced gender ambivalence. These studies give an idea of how nonbinary or agender identities are the general population, but their varying ways of measuring gender identity give a somewhat unclear picture of the prevalence of nonbinary transgender identities. In order to establish proper base rates for how common it is to hold a nonbinary gender identity, further research on population-based samples is needed. To our knowledge, the only Finnish prevalence estimates from population-based samples are those obtained from the above-mentioned adolescent school survey [30], so no estimates from adult populations exist.

## The mental health of sexual minorities

Studies examining the mental health of lesbian, gay, and bisexual individuals have become increasingly methodologically robust in recent years, and many have used population-representative samples. The results of these studies unequivocally indicate that lesbian, gay, and bisexual individuals consistently experience higher levels of depression and anxiety than the heterosexual population [2, 3, 5, 32–34]. Bisexual individuals are especially vulnerable to psychological suffering in comparison to lesbian and gay individuals [32, 33, 35, 36]. Many researchers believe this to be due to double discrimination in the form of stigma experienced both within the LGBTQ community as well as from outside of it [37–39].

Research findings about the mental health of individuals who identify as an EI are in more short supply. Borgogna and colleagues [22] suggested that while all sexual minority participants reported higher levels of anxiety and depression than heterosexual participants, those who identified as EI had the highest rates of anxiety and depression. In accordance with the

minority stress theory, the researchers hypothesized that the poorer mental health outcomes of the EI groups, in comparison to the other sexual minorities, could be attributed to their minority status within the LGBTQ community. Other researchers have reported similar results [40]. Regarding the mental health of asexual individuals, results have varied. People who identified as asexual had more anxiety and depression in comparison to heterosexual individuals in two studies [22, 41], even though others have found no differences [42]. Even though the results of the above-mentioned studies suggest that EI individuals suffer from as much mental health issues as gay, lesbian, and bisexual individuals, more research on this topic is needed.

The vast majority of all studies on the mental health of sexual minority individuals have been conducted on samples from the US and the UK, and few focus on Nordic countries. One of the above-cited studies used a sample from Sweden [32]. There is also one slightly older study from 2009 reporting elevated rates of psychiatric symptoms among adults who report having same-sex sexual interest [43]. Even though their findings did not differ from the pattern observed from other studies, it is still a partly unanswered question whether the mental health outcomes found among sexual minority individuals in these English-speaking countries are similar to the ones in Finland. Furthermore, studies about people who identify as something other than heterosexual, gay, lesbian, or bisexual are especially scarce.

## The mental health of gender minorities

The psychological wellbeing of transgender populations has also been studied actively; however, studies on population-representative samples are few [44]. The majority of studies have focused on binary transgender individuals, and most studies have examined individuals seeking gender-confirming interventions, thereby making generalizations to the overall transgender population difficult [45]. Nonetheless, the existing evidence from systematic literature reviews and meta-analyses strongly suggests that transgender individuals consistently have more mental health problems than cisgender individuals, especially regarding anxiety and depression [4, 44, 46, 47].

Only during the past few years have researchers begun to examine the mental health of those who identify outside the gender binary. In their systematic review on the health of nonbinary individuals, Scandurra and colleagues [48] stated that nonbinary individuals report more negative health outcomes than cisgender participants, though the strength of this conclusion was diminished by the fact that all reviewed studies used convenience samples. From a minority stress perspective, nonbinary individuals may experience greater unique stress and accompanied stigma and discrimination, due to representing a more marginal minority within the LGBTQ community. Moreover, although agender individuals have been recognized in the literature [49, 50], few studies have specifically investigated the mental health of agender individuals. Overall, there is a need for population-representative studies that separately examine the mental health of agender, nonbinary, and binary transgender individuals, in order to gain more reliable evidence about the wellbeing of these minority groups. Furthermore, there is a lack of newer research on the mental health of gender minorities in Finland. Previous studies have found mental health problems to be exceedingly common among transgender youth compared to cisgender peers [51], but no recent studies on Finnish adult samples exist.

## The present study

Although the mental health of sexual and gender minority individuals has been studied actively, there is still a deficiency of empirical evidence confirming whether the mental health disparities detected in smaller, non-probabilistic samples also apply on a population level. Most population-based or population-representative studies on the mental health of people

who identify as LGBTQ have focused on lesbian, gay, and bisexual individuals, with fewer studies focusing on binary transgender individuals. Even fewer studies have examined the mental health of less prevalent sexual and gender minority groups (e.g., pansexual, queer). As mentioned above, there is also a lack of recent studies that analyze adult samples from Finland or other Nordic countries, as most Finnish studies are either older or focus on adolescent samples. Most studies on mental health among sexual and gender minorities explore samples from the US and the UK, and these results may not be generalizable to a Northern European context. Finland as well as the other Nordic countries have long-standing reputations of having relatively high social equality [52], which could hypothetically affect how much minority stress individuals who hold sexual and gender minority identities may experience. The quality of health care in Finland and the other Nordic countries is also high compared to most countries in the world [53], which could hypothetically affect the wellbeing of treatment-seeking transgender individuals. The rights of gender and sexual minorities have also been widely debated in Finland during the last two decades, and several changes in laws and policies have made Finland an even more safe and accommodating country to for sexual or gender minorities [54, 55].

Consequently, our aim with the present study was to provide population-based evidence of the mental health of sexual and gender minorities in Finland, including smaller less well-known minorities (e.g., EI and nonbinary individuals). To our best knowledge, the present study is the first of its kind to analyze anxiety and depression in both sexual and gender minorities in a large population-based sample that also includes smaller and less prevalent identity categories (for similar analyses on non-population-based samples, see [22]). As there is a lack of reliable estimates on how commonly people identify as nonbinary, agender, or EI, we also wanted to explore the prevalence of different sexual orientations and gender identities in the Finnish population.

We hypothesized that:

1. Sexual minority individuals will report higher depression and anxiety symptom scores than heterosexual individuals.

2. Bisexual individuals will report higher depression and anxiety symptom scores than gay and lesbian individuals.

3. Emerging identity individuals and asexual individuals will report higher depression and anxiety symptom scores than gay, lesbian, or bisexual individuals.

4. Minority gender identity individuals (binary transgender, nonbinary, and agender) will report higher depression and anxiety symptom scores than cisgender individuals.

5. Nonbinary and agender individuals will report higher depression and anxiety symptom scores than binary transgender individuals.

6. With regard to minority stress theory, individuals with a double minority status (i.e., those who belong to a sexual minority as well as a gender minority), will report higher depression and anxiety symptom scores than individuals with a single minority status.

## Method

### Participants

The participants were a subset of a population-based sample consisting of Finnish twins and their siblings who participated in an online survey between November 2018 and January 2019. All Finnish twins and siblings of twins who were over 18 years old and currently residing in

Finland were identified from the Finnish Central Population Registry (http://dvv.fi/en). If their mother tongue was listed as Finnish in the Central Population Registry, they were sent an invitation letter by mail to participate in our study. A subset of the invited individuals ($n$ = 7,716) had previously participated in similar data collections by the same research group and had consented to being invited to participate in future studies [56]. Despite some participants having taken part in earlier partly similar surveys conducted by the same research team, the current study was cross-sectional in its design and only data collected in the above-mentioned survey was used for our analyses.

The survey was filled in online with the help of an individual participant code that was provided in the invitation letter. The survey was completely in Finnish and it included questions regarding themes such as sexuality, relationships, and health. A total of 33,211 invitations to participate were sent out, and 9,564 individuals (28.8%) responded by opening the survey with their participant code. Out of the respondents, 9,319 individuals (97.0%) agreed to their data being used for scientific purposes by indicating their written and informed consent via the online survey platform. Out of this group, 8,605 had filled out the anxiety- and depression-related questions relevant for our analyses. Sixteen more participants were omitted due to missing, invalid, or incomprehensible answers about their sexual orientation or gender identity, yielding a final number of 8,589 participants with a mean age of 30.1 years ($Min$ = 18.15 yrs; $Max$ = 60.16 yrs; $SD$ = 8.1).

The majority of the participants (73.3%, $n$ = 6,331) were twins and the rest (26.3%, $n$ = 2,258) were siblings to twins. The ratio of twin participants whose co-twin had also participated was 34.5% of the total sample, yielding 1,482 complete twin pairs. Almost half of all participants were the only members of their family to participate in the study (46.2%, $n$ = 3,965) and were therefore unrelated to other participants. The research project of which the current sample is part of is described in more detail elsewhere, as are the specifics of the data collection used in the present study [56, 57].

## Ethical review

Prior to commencing the data collection, the research plan was reviewed by the Ethics Review Board of Åbo Akademi University. Participation was voluntary and participants could withdraw from the study at any time. The participants provided written informed consent in accordance with the Helsinki Declaration.

## Operationalization of sexual orientation

Participants were asked to describe their sexual orientation using the following alternatives: heterosexual, gay/lesbian, bisexual, and other. Participants who chose the last alternative could describe their orientation in a textbox. We manually reviewed these textbox answers, omitted participants with missing or uninterpretable answers (e.g., "mystery" and "perv"), and grouped the rest into categories. Details about the categorization procedure are found in the Online Supplement (S1 File).

The distribution of sexual orientations and gender identities is illustrated in Table 1. We will refer to pansexual, questioning/undefined, demisexual, and queer as EI, and in the statistical analyses the EI groups were combined and analyzed as a whole due to small sample sizes. Another word that has been used as an umbrella term for many types of non-normative sexual minority identities and gender identities is queer, but as some of the people in this group specifically identified as queer ($n$ = 4) and most others ($n$ = 96) reported other terms (e.g., pansexual, demisexual), we deemed it inaccurate to refer to the whole group using an umbrella term that is synonymous with what part (but not all) of this subsample described themselves as.

**Table 1. The frequency of different sexual orientations and gender identities among the participants.**

| Sexual Orientation | Gender Identity | | | | | |
|---|---|---|---|---|---|---|
| | Cisgender | Transgender man | Transgender woman | Nonbinary | Agender | Total |
| Heterosexual | 7,562 | 2 | 2 | 0 | 4 | 7,570 |
| Gay/Lesbian | 220 | 4 | 0 | 7 | 0 | 231 |
| Bisexual | 597 | 8 | 5 | 10 | 6 | 626 |
| Pansexual | 47 | 2 | 0 | 5 | 4 | 58 |
| Asexual | 46 | 3 | 0 | 10 | 3 | 62 |
| Questioning/ Undefined | 28 | 0 | 0 | 0 | 0 | 28 |
| Demisexual | 8 | 0 | 0 | 2 | 0 | 10 |
| Queer | 3 | 0 | 0 | 1 | 0 | 4 |
| Total | 8,511 | 19 | 7 | 35 | 17 | 8,589 |

Although asexual individuals are sometimes included under the EI umbrella, they were analyzed separately since our sample size was sufficient to allow for this.

## Operationalization of gender identity

The participants were first presented with the statement "According to the population registry your gender is male/female" (depending on the participant's gender as listed in the Central Population Registry in Finland) and given two response options; (a) yes, this is correct and in line with my gender identity; or (b) no, I do not identify as male/female. We then asked a follow-up question "Which of the following alternatives describes your gender identity best?", to which the response options depended on the gender stated in the Central Population Registry. For individuals listed as male, the alternatives were man, transgender man, transgender woman, and other. For individuals listed as female, the options were woman, transgender woman, transgender man, and other. If the participant chose the alternative "other", they could describe their identity in a textbox. We reviewed and categorized the free-text answers into more descriptive categories, omitting participants with missing, uninterpretable, or inadequate answers (e.g., "attack helicopter" and "human"). Details about the categorization procedure are found in the Online Supplement (S1 File). The majority of the participants who reported that their gender identity aligned with the gender reported in the Central Population Registry were women (65.9%), whereas the rest were men (34.1%),

In the present study, we chose to use the term nonbinary to describe all identities outside or between the gender binary. Even though agender is often included among the nonbinary identities, we chose to regard them as a separate group for the purpose of this study, as agender specifically refers to the absence of gender identity and can also be considered as a category that is different from other nonbinary identities.

After the categorization, the final gender identity categories were cisgender ($n$ = 8,511), transgender man ($n$ = 19), transgender woman ($n$ = 7), nonbinary ($n$ = 35), and agender ($n$ = 17). Due to small sample sizes, the transgender men and transgender women were combined into one category of binary transgender participants for the statistical analyses.

## Measures

Anxiety and depression were measured with the six-item anxiety and depression subscales from the Brief Symptom Inventory-18 [58], which is a self-report inventory using a 5-point Likert-type scale (1 = Not at all; 2 = A little bit; 3 = Moderately; 4 = Quite a bit; 5 = Extremely). Both subscales were made into separate composite variables consisting of the sum of the

individual items belonging to each subscale. Higher scores indicated more symptoms (range 6–30). Both the depression (Cronbach's α = .87) and the anxiety (Cronbach's α = .88) measures displayed good internal consistency. The original English BSI-18 items were translated into Finnish by a researcher with an academic degree from a British university and relevant expertise on mental health, and the accuracy of the translation was checked by using a back-translation procedure.

### Statistical analyses

We used IBM SPSS Statistics 26.0 for Windows and Generalized Estimating Equations (GEE) multilevel regression modeling to examine differences in depression and anxiety between the sexual orientation and gender identity groups. We chose GEE because it allows for controlling for between-subjects dependence arising due to the genetic relatedness of the subjects. We coded all participants from the same family with the same code, so that all twins and siblings who were interrelated had the same value on the family variable. We then added the family variable as a subject variable in the GEE analyses. Age was included as a covariate in all analyses.

To investigate the first hypothesis, we conducted a GEE analysis comparing heterosexual ($n = 7,570$) and sexual minority participants (all minority categories; $n = 1,019$). To investigate the second and third hypotheses, we used another more differentiated sexual orientation variable, with the categories heterosexual ($n = 7,570$), gay/lesbian ($n = 231$), bisexual ($n = 626$), EI ($n = 100$), and asexual ($n = 62$). This variable was created to compare the minorities with each other, as well as to heterosexual individuals. When investigating the fourth hypothesis, we used a gender identity variable, with the categories cisgender ($n = 8,511$) and gender minority ($n = 78$). This variable was created to analyze differences between gender majority and minority individuals. To investigate the fifth hypothesis, we used a gender identity variable with the categories cisgender ($n = 8,511$), binary transgender ($n = 26$), nonbinary ($n = 35$), and agender ($n = 17$). This variable was created to compare the gender minorities with each other, as well as to cisgender individuals. Lastly, to study the differences in depression and anxiety according to minority status, we created a new minority status variable with three levels; 1 = no minority status ($n = 7,562$), 2 = single minority status ($n = 957$), and 3 = double minority status ($n = 70$). To investigate the sixth hypothesis, this newly created minority status variable was used to explore if individuals with a double minority status would report higher levels of anxiety and depression.

The number of unique comparisons we made in our analyses amounted to 32, so we chose to correct for multiple testing using the Bonferroni method where alpha values are divided by the number of comparisons in order to control for type I error. The corrected alpha level was 0.05 / 32 = .0015. The syntax for the analyses can be found in the Online Supplement (S2 File).

## Results

In our sample, 88.13% identified as heterosexual, 2.68% as gay/lesbian, and 7.28% as bisexual. The remaining 1.16% identified as part of the EI (here including asexual individuals), of which 0.67% identified as pansexual, 0.72% as asexual, 0.32% as questioning/undefined, 0.11% as demisexual, and 0.04% as queer. Regarding gender identity, 99.09% of our sample identified as cisgender. Of the 0.91% of individuals who identified as non-cisgender, 0.30% identified as binary transgender, 0.40% as nonbinary, and 0.20% as agender.

### Sexual orientation, depression and anxiety

Sexual minority participants reported significantly more depressive symptoms ($M = 14.20$, $SE = 0.19$) than heterosexual participants ($M = 11.97$, $SE = 0.06$). Estimated marginal means

for the groups can be seen in Table 2 and the results from the GEE analyses comparing sexual minorities to heterosexual individuals can be seen in Table 3. Sexual minority participants also reported significantly higher rates of anxiety ($M$ = 12.63, $SE$ = 0.18) than heterosexual individuals ($M$ = 10.59, $SE$ = 0.06), thereby supporting the first hypothesis. Age had a significant negative association with depression (Wald $\chi^2$ = 253,457, $\beta$ = -.111, $SE$ = 0.0070, 95% CI [-1.25, -0.97], $p$ < .001) and anxiety (Wald $\chi^2$ = 181,822, $\beta$ = -.083, $SE$ = 0.0062, 95% CI [-0.95, -0.71], $p$ < .001), indicating that higher age was associated to lower rates of depressive and anxious symptoms and vice versa. When comparing the depression symptom scores of the sexual minority groups separately to the rates of heterosexual individuals, the differences between heterosexual participants and the different sexual minority subgroups were nominally significant for all groups, indicating that all sexual minority groups reported higher depression symptom scores than heterosexual participants. When applying the Bonferroni corrected alpha level to the results, the differences remained significant for the EI and the bisexual groups. When comparing the rates of anxiety of the sexual minority groups separately to the rates of heterosexual participants, asexual individuals were the only group that did not report significantly higher rates of anxiety ($M$ = 11.64, $SE$ = 0.74).

Bisexual individuals reported significantly higher levels of depression ($M$ = 14.63, $SE$ = 0.24) than gay and lesbian participants ($M$ = 12.81, $SE$ = 0.35). See Tables 4 and 5 for means and mean differences between the sexual minority groups on depression and anxiety scores. Bisexual participants also reported higher rates of anxiety ($M$ = 12.97, $SE$ = 0.23) than gay and lesbian participants ($M$ = 11.73, $SE$ = 0.34). The difference in anxiety was nominally significant and did not remain so when applying the Bonferroni correction. Our second hypothesis therefore received only partial support. The EI group reported significantly higher depression symptom scores ($M$ = 15.01, $SE$ = 0.59) than gay and lesbian participants. The anxiety ($M$ = 13.20, $SE$ = 0.63) symptom scores that were nominally higher for the EI group than the gay and lesbian group, but the difference was not robust enough to be significant when

**Table 2. Estimated marginal means for depression and anxiety.**

| Participant group | | | N | Depression | | | Anxiety | | |
|---|---|---|---|---|---|---|---|---|---|
| | | | | M | SE | Range | M | SE | Range |
| Sexual Orientation | | | | | | | | | |
| | Heterosexual | | 7,570 | 11.97 | 0.06 | 6–30 | 10.59 | 0.06 | 6–30 |
| | Sexual Minority | | 1,019 | 14.20 | 0.19 | 6–30 | 12.63 | 0.18 | 6–30 |
| | | Gay/-Lesbian | 231 | 12.81 | 0.35 | 6–29 | 11.73 | 0.34 | 6–30 |
| | | Bisexual | 626 | 14.63 | 0.24 | 6–30 | 12.97 | 0.23 | 6–30 |
| | | Emerging Identity | 100 | 15.01 | 0.59 | 6–30 | 13.20 | 0.63 | 6–29 |
| | | Asexual | 62 | 13.83 | 0.78 | 6–27 | 11.64 | 0.74 | 6–30 |
| Gender identity | | | | | | | | | |
| | Cisgender | | 8,511 | 12.21 | 0.06 | 6–30 | 10.80 | 0.05 | 6–30 |
| | Gender Minority | | 78 | 14.95 | 0.65 | 6–30 | 13.50 | 0.66 | 6–29 |
| | | Binary Transgender | 26 | 13.73 | 1.09 | 6–26 | 11.79 | 0.87 | 6–25 |
| | | Nonbinary | 35 | 16.50 | 0.91 | 8–30 | 15.46 | 1.03 | 6–29 |
| | | Agender | 17 | 13.64 | 1.49 | 6–28 | 12.09 | 1.39 | 6–26 |
| Single Minority Status | | | 957 | 14.15 | 0.19 | 6–30 | 12.56 | 0.19 | 6–30 |
| Double Minority Status | | | 70 | 14.94 | 0.67 | 6–30 | 13.59 | 0.69 | 6–29 |

*Note*. The means are estimated marginal means from the Generalized Estimating Equations analysis. $M$ = mean, $SE$ = standard error, Range = minimum–maximum of the range.

**Table 3. Comparing sexual minorities to heterosexual individuals on anxiety and depression.**

| Variable & group | | | Wald χ² (1) | β | 95% CI | SE | p |
|---|---|---|---|---|---|---|---|
| Depression | | | | | | | |
| | Sexual Minorities Collectively | | 132.83 | 2.23 | [1.85, 2.61] | 0.19 | < .001 |
| | | Gay/-Lesbian | 5.71 | 0.84 | [0.15, 1.53] | 0.35 | .017 |
| | | Bisexual | 120.27 | 2.66 | [2.19, 3.14] | 0.24 | < .001 |
| | | Emerging Identity | 26.61 | 3.05 | [1.89, 4.20] | 0.59 | < .001 |
| | | Asexual | 5.75 | 1.87 | [0.34, 3.39] | 0.78 | .016 |
| Anxiety | | | | | | | |
| | Sexual Minorities Collectively | | 117.25 | 2.04 | [1.67, 2.41] | 0.19 | < .001 |
| | | Gay/ Lesbian | 11.25 | 1.14 | [0.48, 1.81] | 0.34 | < .001 |
| | | Bisexual | 102.54 | 2.40 | [1.92, 2.85] | 0.24 | < .001 |
| | | Emerging Identity | 17.13 | 2.61 | [1.38, 3.85] | 0.63 | < .001 |
| | | Asexual | 2.03 | 1.06 | [-0.40, 2.51] | 0.74 | .154 |

*Note*. Results from Generalized Estimating Equations Analysis. Heterosexual participants were the comparison group in all comparisons. *SE* = standard error.
CI = confidence interval.

applying the Bonferroni correction. This left our third hypothesis with partial support. No significant differences in rates of depression were found between the gay and lesbian group and the asexual group (*M* = 13.83, *SE* = 0.78). Likewise, no significant differences were found in rates of anxiety between the gay and lesbian group and the asexual group (*M* = 11.64, *SE* = 0.74). There were also no significant differences between bisexual, EI, and asexual individuals in rates of either depression or anxiety. Age had a significant negative association with

**Table 4. Comparisons of depression symptom scores by sexual orientation.**

| Comparison group (A) | Other orientations (B) | Mean difference (A—B) | SE | P | 95% CI |
|---|---|---|---|---|---|
| Gay/ Lesbian | Heterosexual | 0.84 | 0.35 | .017* | [0.15, 1.53] |
| | Bisexual | -1.82 | 0.42 | < .001*** | [-2.64, -1.00] |
| | Emerging Identity | -2.20 | 0.68 | .001** | [-3.54, -0.87] |
| | Asexual | -1.02 | 0.84 | .221 | [-2.67, 0.62] |
| Bisexual | Heterosexual | 2.66 | 0.24 | < .001*** | [2.19, 3.14] |
| | Gay/ lesbian | 1.82 | 0.42 | < .001*** | [1.00, 2.64] |
| | Emerging Identity | -0.39 | 0.63 | .540 | [-1.62, 0.85] |
| | Asexual | 0.79 | 0.81 | .325 | [-0.79, 2.37] |
| Emerging Identity | Heterosexual | 3.05 | 0.59 | < .001*** | [1.89, 4.20] |
| | Gay/ lesbian | 2.20 | 0.68 | .001** | [0.87, 3.54] |
| | Bisexual | 0.39 | 0.63 | .540 | [-0.85, 1.62] |
| | Asexual | 1.18 | 0.98 | .228 | [-0.74, 3.10] |
| Asexual | Heterosexual | 1.87 | 0.78 | .016* | [0.34, 3.39] |
| | Gay/ lesbian | 1.02 | 0.84 | .221 | [-0.62, 2.67] |
| | Bisexual | -0.79 | 0.81 | .325 | [-2.37, 0.79] |
| | Emerging Identity | -1.18 | 0.98 | .228 | [-3.10, 0.74] |

*Note*. Pairwise comparisons of mean differences in depression rates from the Generalized Estimating Equations analysis. *SE* = standard error. CI = confidence interval.
* *p* < .05.
** *p* < .01.
*** *p* < .001

**Table 5. Comparisons of anxiety symptom scores by sexual orientation.**

| Comparison group (A) | Other minorities (B) | Mean difference (A—B) | SE | P | 95% CI |
|---|---|---|---|---|---|
| Gay/ Lesbian | Heterosexual | 1.14 | 0.34 | < .001*** | [0.48, 1.81] |
| | Bisexual | -1.24 | 0.41 | .002** | [-2.04, -0.45] |
| | Emerging Identity | -1.47 | 0.71 | .039* | [-2.87, - 0.08] |
| | Asexual | 0.09 | 0.80 | .915 | [-1.48, 1.66] |
| Bisexual | Heterosexual | 2.39 | 0.24 | < .001*** | [1.92, 2.85] |
| | Gay/ lesbian | 1.24 | 0.41 | .002** | [0.45, 2.04] |
| | Emerging Identity | -0.23 | 0.66 | .733 | [-1.53, 1.07] |
| | Asexual | 1.33 | 0.77 | .085 | [-0.19, 2.85] |
| Emerging Identity | Heterosexual | 2.61 | 0.63 | < .001*** | [1.38, 3.85] |
| | Gay/ lesbian | 1.47 | 0.71 | .039* | [0.08, 2.87] |
| | Bisexual | 0.23 | 0.66 | .733 | [-1.07, 1.53] |
| | Asexual | 1.56 | 0.97 | .110 | [-0.35, 3.47] |
| Asexual | Heterosexual | 1.06 | 0.74 | .154 | [-0.40, 2.51] |
| | Gay/ lesbian | -0.09 | 0.80 | .915 | [-1.66, 1.48] |
| | Bisexual | -1.33 | 0.77 | .085 | [-2.85, 0.19] |
| | Emerging Identity | -1.56 | 0.97 | .110 | [-3.47, 0.35] |

*Note.* Pairwise comparisons of mean differences in anxiety rates from the Generalized Estimating Equations analysis. *SE* = standard error. CI = confidence interval.

* *p* < .05.

** *p* < .01.

*** *p* < .001

anxiety (Wald $\chi^2$ = 248,186, β = -.082, *SE* = 0.0070, 95% CI [-1.23, -0.96], *p* < .001) and depression (Wald $\chi^2$ = 178,685, β = -.110, *SE* = 0.0062, 95% CI [-0.94, -0.70], *p* < .001) in the analyses where we compared the different sexual minority groups against each other.

## Gender identity, depression, and anxiety

Gender minority participants reported significantly higher rates of depression (*M* = 14.95, *SE* = 0.65) than cisgender participants (*M* = 12.21, *SE* = 0.06), as well as significantly higher rates of anxiety (*M* = 13.50, *SE* = 0.66) than cisgender participants (*M* = 10.80, *SE* = 0.05), thereby supporting the fourth hypothesis. Age had a significant negative association with depression (Wald $\chi^2$ = 302,941, β = -.122, *SE* = 0.0070, 95% CI [-0.135, -0.108], *p* < .001) and anxiety (Wald $\chi^2$ = 223,051, β = -.093, *SE* = 0.0062, 95% CI [-0.105, -0.081], *p* < .001) when comparing cisgender participants to gender minority participants in general. When comparing the results of each minority group separately to the cisgender individuals, only nonbinary individuals had significantly higher rates of depression (*M* = 16.50, *SE* = 0.91) in comparison to cisgender participants. Nonbinary individuals were also the only gender minority to report higher rates of anxiety (*M* = 15.46, *SE* = 1.03) than cisgender individuals. See Table 6 for results from the GEE analyses comparing gender minorities to cisgender individuals.

There was no difference in depression symptom scores between nonbinary (*M* = 16.50, *SE* = 0.91) and binary transgender individuals (*M* = 13.73, *SE* = 1.09) (Please see S1 Table for more detailed results). The difference between nonbinary (*M* = 15.46, *SE* = 1.03) and binary transgender individuals (*M* = 11.79, *SE* = 0.87) on anxiety symptom scores was nominally significant with nonbinary individuals experiencing more anxiety, but the difference was not significant after controlling for multiple tests. We found no significant differences in depression scores between binary transgender and agender participants (*M* = 13.64, *SE* = 1.49). Similarly,

**Table 6. Comparing gender minority individuals to cisgender individuals on anxiety and depression.**

| | | | Wald χ² (1) | β | 95% CI | SE | P |
|---|---|---|---|---|---|---|---|
| Depression | | | | | | | |
| | Gender Minorities Collectively | | 17.45 | 2.75 | [1.46, 4.04] | 0.66 | < .001*** |
| | | Binary Transgender | 1.96 | 1.53 | [-0.6, 3.66] | 1.09 | .161 |
| | | Nonbinary | 22.20 | 4.29 | [2.51, 6.08] | 0.91 | < .001*** |
| | | Agender | 0.92 | 1.43 | [-1.50, 4.36] | 1.49 | .338 |
| Anxiety | | | | | | | |
| | Gender Minorities Collectively | | 16.86 | 2.70 | [1.41, 4.00] | 0.66 | < .001*** |
| | | Binary Transgender | 1.28 | 0.99 | [-0.73, 2.70] | 0.87 | .259 |
| | | Nonbinary | 20.56 | 4.66 | [2.65, 6.67] | 1.03 | < .001*** |
| | | Agender | 0.86 | 1.29 | [-1.43, 4.01] | 1.39 | .353 |

*Note.* The results are from a Generalized Estimating Equations analysis. Cisgender participants were the comparison group in all analyses. *SE* = standard error. CI = confidence interval.

* *p* < .05.

** *p* < .01.

*** *p* < .001

there were no significant differences in anxiety scores between binary transgender and agender participants (*M* = 12.09, *SE* = 1.39), thereby partially contradicting the fifth hypothesis. Nonbinary individuals experienced higher rates of anxiety (*M* = 15.46, *SE* = 1.03), in comparison to agender individuals (*M* = 12.09, *SE* = 1.39), but the result did not remain significant after controlling for multiple tests. We once again detected a significant negative association between age and anxiety (Wald χ² = 224,058, β = -.093, *SE* = 0.0062, 95% CI [-.11, -0.81], *p* < .001) and between age and depression (Wald χ² = 303,553, β = -.122, *SE* = 0.0070, 95% CI [-0.14, -0.11], *p* < .001) when comparing the different gender minority groups with each other. See S1 Table for more detailed results from the pairwise comparisons of mean differences between the gender minorities on depression and anxiety scores.

### Minority status, depression, and anxiety

When comparing individuals with a double minority status (*M* = 14.95, *SE* = 0.67) to those with a single minority status (*M* = 14.16, *SE* = 0.19), no significant differences in rates of depression were found (*Mean difference* = 0.79, *SE* = 0.70, 95% CI [-0.58–2.16], *p* = .260). Likewise, there was no significant difference between participants with a double minority status (*M* = 13.60, *SE* = 0.69) and participants with a single minority status (*M* = 12.56, *SE* = 0.19) on anxiety scores (*Mean difference* = 1.03, *SE* = 0.71, 95% CI [-0.36–2.43], *p* = .145). There was a negative association between age and anxiety (Wald χ² = 180,322, β = -.083, *SE* = 0.0062, 95% CI [-.095, -.071], *p* < .001) and between age and depression (Wald χ² = 251,798, β = -.111, *SE* = .0070, 95% CI [-.124, -.097], *p* < .001) in these analyses, as well.

### Discussion

The present study was, to our knowledge, the first of its kind to use a population-based sample to examine differences in anxiety and depression both between the gender and sexual minorities as well as compared to the heterosexual cisgender majority. Based on minority stress theory [7–9] and previous findings [2, 22, 34, 59], we expected that sexual and gender minority individuals would report higher rates of depression and anxiety than majority individuals, and that there would also be differences between the minority groups. Based on and previous

preliminary results [15] we also expected individuals with a double minority status to experience more anxiety and depression than those with a single minority status.

## Main findings and interpretations

Of our sample, 88.13% identified as heterosexual, 2.68% as gay/lesbian, and 7.28% as bisexual. This prevalence estimate of 9.96% of individuals in total identifying as lesbian, gay, or bisexual in our sample is somewhat higher than previous estimations [20]. The remaining 1.16% of individuals identified as part of the EI (including asexual individuals). Of these participants, 0.67% were pansexual, 0.72% were asexual, 0.32% were questioning/undefined, 0.11% were demisexual, and 0.04% were queer. These estimates align with the other existing population-based prevalence estimates [26, 27].

Regarding gender identity, 99.09% of the participants identified as cisgender and 0.91% as transgender. Of the transgender participants, 0.30% were binary transgender, 0.40% were non-binary, and 0.20% were agender. These estimates are similar to the ones obtained by Gates [20]. However, our sample had notably lower prevalence rates than the ones obtained in Belgium by van Caenegem and colleagues [31]. The vast prevalence differences are probably caused by the fact that van Caenegem and colleagues measured experiences of gender incongruence instead of gender identity, as this may allow participants to report experiences of gender incongruence without endorsing a specific gender identity label. At least in research regarding sexual orientation, the proportion of participants reporting same-sex sexual attraction has been shown to be vastly greater than the proportion of participants who self-identify as non-heterosexual [60].

In accordance with our hypotheses, sexual and gender minority individuals experienced significantly more anxiety and depression than cisgender and heterosexual individuals (hypotheses 1 and 4). Our results indicate that even though Finland is a developed country with a relatively well-functioning health care system [53, 61] and a relatively inclusive social climate [52, 62], there is still progress to be made as sexual and gender minorities are disproportionally affected by mental health issues.

Our results also imply that certain minorities within the LGBTQ umbrella are especially vulnerable to experiencing anxiety and depression. In accordance with our second hypothesis, bisexual individuals experienced more depression than heterosexual, lesbian, and gay individuals. There was a nominally significant difference for anxiety, as well, but it did not hold up when correcting for multiple tests. Individuals who identified as part of an EI experienced higher rates of depressive symptoms and nominally higher rates of anxiety symptoms compared to the lesbian and gay participants, but there were no differences compared to the bisexual participant group. Our third hypothesis about EI individuals being more vulnerable to depression and anxiety therefore received partial support.

Our findings also suggest that individuals with a nonbinary gender identity have slightly more anxiety than binary transgender individuals and slightly more depression and anxiety than agender individuals, but these results were not robust enough to remain significant after controlling for multiple tests. No differences were detected between agender and binary transgender participants for either depression or anxiety. Therefore, our fifth hypothesis only received partial and very tentative support.

The above-mentioned discrepancies in mental health outcomes between minorities are partially consistent with previous findings [15, 22, 40]. As EI and nonbinary individuals constitute minorities within the LGBTQ community, they may experience additional minority stress from both within and outside the community, which could partly explain the partially elevated depression and anxiety levels. Belonging to a minority within a minority could potentially

contribute to increased minority stress, as one may face more widespread ignorance concerning one's identity, as well as more discrimination, which ultimately could result in poorer mental health [48]. In other words, being EI or nonbinary could be seen a form of elevated minority status. On the other hand, bisexual individuals were the largest minority group of all sexual minorities, but our results indicate that they experience higher levels of anxiety and depression than many other LGBTQ minority groups. These findings are in line with previous evidence suggesting that bisexual individuals suffer more from mental health issues than lesbian and gay individuals (see e.g. [33]). The higher levels of psychopathology among bisexual individuals are thought to be a result of social stress due to negative social attitudes such as monosexism (i.e., the belief that individuals can only be heterosexual, gay, or lesbian), as well as discrimination from both within and outside the LGBTQ community [37–39].

Another interesting result in the present study was that we found no significant differences in depression or anxiety between the cisgender and the binary transgender group or between the cisgender and agender group. This was surprising and contradictory to previous findings suggesting that binary transgender individuals suffer from more mental health problems than cisgender individuals (see e.g., [4, 22]). Our findings could result from our methodology and the sample used in the current study. Unlike many previous studies (see e.g., [44]), we did not use LGBTQ-targeted sampling, the sample was not collected in a clinical setting, nor was the survey advertised as a LGBTQ health study. In conclusion, due to the general lack of research on non-treatment-seeking transgender individuals in the general population, it is also possible that this group does not suffer from depression and anxiety as much as previous evidence has suggested. Future research is needed to establish how frequent mental health issues are in the binary transgender and agender population overall, and whether they in fact are more prevalent when analyzing non-clinical population-representative samples and making comparisons to the cisgender population.

On the other hand, our results indicate that nonbinary individuals suffer more from anxiety and depression than cisgendered individuals but not significantly more than binary transgender individuals, which is not entirely in line with previous findings [15, 22, 63]. This should be taken into consideration in health care services in general as well as in specific gender minority-targeted services and interventions, as nonbinary individuals are a minority group which is less known and therefore probably less well understood by mental health service providers.

Regarding the asexual group, we found that individuals who identify as asexual do not experience significantly more depression or anxiety compared to heterosexual individuals after having controlled for multiple comparisons in our analyses. These results are not entirely in line with most previous findings suggesting higher rates of mental health problems in asexual individuals [22, 41]. As previous studies have mostly used different non-probability sampling methods (see e.g. [41]), the results of the present study are a welcome contribution to this literature.

Contrary to previous findings [22], our results suggest that individuals belonging to both a sexual and gender minority (double minority status) do not suffer from significantly higher rates of anxiety or depression, in comparison to those identifying as either a sexual or gender minority (single minority status). These non-significant results may stem from the small sample size of the double minority status group. Another way of understanding our results is in the light of the hypothesis that groups with an elevated minority status could also develop greater resilience towards additional kinds of discrimination, and as a result they might be better prepared to cope with discrimination on the whole [64]. Therefore, it could also be possible that holding both a gender and a sexual minority identity does not have additional impact secondary to holding one gender or sexual minority status.

The lack of differences between the single and double minority groups may also reflect the fact that our operationalization of double minority status is a narrow one that excludes many

other possible types of double minorities that may impact mental health. For example, studies on US populations have found that Black women who belong to sexual minorities experience poorer mental health compared to both white sexual minority women and black men who belong to sexual minorities [65]. Another example of elevated minority status and its associations to poorer mental health can be seen in how sexual and gender minority individuals who also present with autism report poorer mental health compared to heteronormative populations [66]. Not only did individuals with both autism and gender dysphoric traits experience more mental health problems, but autism coupled with a non-heterosexual orientation did not associate with poorer mental health unless gender dysphoric traits were also present. In other words, our operationalization of double minority status is only one way of trying to compare how elevated minority status may be associated with more anxiety and depression, and our findings may result either from a lack of power or from other, presently unmeasured variables that may also affect mental health. As mentioned above, being part of a minority within a minority (e.g., EI or nonbinary) could also be seen as a type of elevated minority status that may even be comparable to holding several intersecting minority identities. In other words, measuring and quantifying minority status or one's degree of marginalization is no easy task. More empirically and methodologically robust research in this field is nevertheless needed to explore the effects of multiple minority statuses on mental health.

## Strengths and limitations

Declining survey response rates is a problem that affects all psychological research, and this seems also true for the present study with its response rate of about 29% [67]. However, as the questionnaire from which the data in the present study was extracted was extensive and featured a number of questions on sensitive topics such as sex and drug use, the response rate must still be considered decent [68]. Furthermore, results concerning sexuality-related topics in our sample were similar to other Finnish population-based studies with higher response rates (e.g. frequency of masturbation, [69]).

We did not ask for information about participants' race/ethnicity, income or education level. Asking for this type of demographic information would have helped us to further clarify how well our sample represents the Finnish population on average, so omitting these types of questions can be seen as a limitation to our study. However, Finland is a very ethnically/racially homogenous country, especially when looking at the proportion of Finnish citizens who have Finnish as their native language [70]. Finland and the rest of the Nordic countries also stand out with their easily accessible and free education, which likely decreases inequality and segregation. The proportion of people with a post-comprehensive education is high in international comparison, as is the proportion of people who have a tertiary (i.e., university-level) education [71]. Differences in income are also small in comparison to most countries [72].

Despite the fact that we had a relatively large sample at our disposal, some minority groups were too small to further divide by gender. The groups of heterosexual, gay/lesbian, and bisexual participants we large enough to be divided, but as this was not the case for the EI and asexual participants, dividing some groups but not others would have led to us comparing single-sex groups (e.g., bisexual women) and mixed groups (EI individuals of all genders) to each other within the same analyses. We therefore chose to not separate any of the groups by gender.

The small sample sizes of some of the minority groups did not allow for the EI groups to be analyzed individually, and it may also have contributed to some of the non-significant results by making our analyses underpowered. Exploring infrequent phenomena in a population-based sample is associated with inevitably large group size differences and subsequent

analytical challenges, even though population-based samples have clear inherent advantages such as decreased risk for participant-selection bias and increased generalizability.

Another similar limitation is that binary transgender women and men could not be separated in the analyses due to small sample sizes, making it impossible to compare possible gender differences. Future population-based research should, consequently, aim to separate binary transgender men and women in order to explore these differences in more detail. The level of psychopathology experienced by transgender individuals seems to subside following gender-affirming interventions, which means that gender affirmative treatment probably affects one's psychological wellbeing in a profound way [44, 73, 74]. It would therefore be helpful to explore mental health in samples where it is possible to take into account whether the person has been able to or has chosen to seek gender affirmative treatment. One's experiences of the effects of any obtained gender affirmative treatment could be another future variable of interest, as well, as the outcome of the gender affirmative interventions could logically be associated with subsequent psychological wellbeing.

Despite the above-mentioned limitations, the present study is the first of its kind in Finland to explore the mental health of nonbinary and EI individuals using a large population-based sample. Another strength was that the participants could self-identify as sexual and gender minorities, as self-identification is preferable to measures of sexual behavior and gives a more accurate reflection of experienced identity. On the other hand, self-identification and write-in answers could also be considered a limitation, as some answers might have been misinterpreted or categorized erroneously. For example, individuals who stated two identities were classified according to the first identity mentioned, and may therefore have been labeled in a way that emphasized the less prominent identity instead of the more prominent one. Self-identification via endorsement of identity labels also has the limitation of not including people who experience gender incongruence or non-heterosexual interest but do not self-identify with any particular label signifying a non-normative sexual or gender identity.

As a final point of discussion, we would like to mention the use of corrections for multiple testing, which we used in order to control the increased risk for Type I error that follows from running a high number of statistical comparisons. The Bonferroni correction we used is a very stringent method of assuring that the results are indicative of true effects, but at the same time it is important to bear in mind that a correction method this strict also inflates the probability of true effects going undetected. In the case of the differences in mental health between EI individuals and gay, lesbian, and bisexual individuals, the nominally significant results we received were in line with previous research even though they did not hold up to the corrected alpha levels.

## Conclusions

Our results align with previous findings indicating that sexual and gender minority individuals overall suffer from higher levels of anxiety and depression than the cisgender heterosexual majority. According to our results, bisexual, EI, and nonbinary individuals are especially vulnerable to anxiety and depression. Considering the debilitating nature of both depression and anxiety disorders, acknowledging and addressing these discrepancies in mental health is an important task both on the level of clinical psychology and health care policy. Additional research is, however, needed to gain more in-depth knowledge about these mental health discrepancies and their effect on minority individuals. Future studies should aim to examine the effects of multiple minority statuses on mental health, as well as examine in particular the psychological wellbeing of the emerging identity groups, in order to gain more specific information about the separate subgroups.

## Supporting information

**S1 File. Categorization of gender identity and sexual orientation.** Detailed information on the categorization of textbox responses on the sexual orientation and gender identity questions.
(DOCX)

**S2 File. SPSS syntax for the GEE analyses.**
(DOCX)

**S1 Table. Comparison of anxiety and depression symptom scores by gender identity.**
(DOCX)

## Acknowledgments

We would like to thank Christa Bäckström, Martin Lagerström, and Amy Lindroos for their valuable comments on the manuscript.

## Author Contributions

**Conceptualization:** Marianne Källström, Annika Gunst.

**Data curation:** Marianne Källström, Annika Gunst.

**Formal analysis:** Marianne Källström, Nicole Nousiainen, Annika Gunst.

**Funding acquisition:** Patrick Jern.

**Investigation:** Patrick Jern, Annika Gunst.

**Methodology:** Patrick Jern, Sabina Nickull, Annika Gunst.

**Project administration:** Marianne Källström, Annika Gunst.

**Resources:** Patrick Jern.

**Supervision:** Marianne Källström, Patrick Jern, Annika Gunst.

**Visualization:** Sabina Nickull.

**Writing – original draft:** Marianne Källström, Nicole Nousiainen.

**Writing – review & editing:** Marianne Källström, Nicole Nousiainen, Patrick Jern, Sabina Nickull, Annika Gunst.

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
