## [Decision Letter · Decision Letter 0]

22 Jun 2022

PONE-D-22-08353Mental health among sexual and gender minorities: A Finnish population-based study of anxiety and depression discrepancies between individuals of diverse sexual orientations and gender minorities and the majority populationPLOS ONE

Dear Dr. Källström,

Thank you for submitting your manuscript to PLOS ONE. After careful consideration, we feel that it has merit but does not fully meet PLOS ONE’s publication criteria as it currently stands. Therefore, we invite you to submit a revised version of the manuscript that addresses the points raised during the review process.

We look forward to receiving your revised manuscript.

Kind regards,

Michelle Torok, Ph.D.

Academic Editor

PLOS ONE

Journal Requirements:

Additional Editor Comments:

This paper, which reports on the burden of depression and anxiety among sexual and gender minorities, is overall well-written, with good theoretical justification, and reports on an important issue. In addition to the valid concerns raised by the three reviewers, I note some issues to be addressed.

1. In respect to the hypotheses, I feel these could be more clearly operationalised. For example, in hypothesis 5, instead of saying “….will report more depression and anxiety” it would be more accurate to state “….will report higher symptom scores” or similar. With respect to the other hypotheses, it would be better state that you are measuring anxiety and depression symptoms, rather than ‘higher levels of depression and anxiety’.

2. Is the survey cross-sectional or longitudinal? Please state this in the Methods. Was informed consent provided online, in person, or pen-and-paper? How were the participants recruited?

3. Out of interest, did the survey capture how long/for what duration individuals had identified as a particular gender or sexuality? One would expect higher levels of depression and/or anxiety symptoms for those who had recently chosen to identify in a particular way versus those who were more settled into their identity.

4. Did the survey capture any information about why participants may or may not be experiencing mental ill-health? (e.g., trauma, violence, bullying etc?). If so, I would suggest including these in your analysis via regression modelling to determine if gender and sexuality minority remained independently associated with depression and anxiety symptoms. This information would help build a fuller discussion, which is currently full of hypothesizing about the stress load of minority groups as the authors have not looked at potential explanatory factors.

5. As per other reviewers’ comments, please be more explicit in reporting age adjustments in the results section.

6. Tables 7 and 8 don’t appear to add a lot to the paper. At the very least, I would suggest removing them to a supplementary materials section and briefly reporting them in text if the authors feel this information is needed.

Reviewers' comments:

Reviewer's Responses to Questions

**Comments to the Author**

1. Is the manuscript technically sound, and do the data support the conclusions?

Reviewer #1: Yes

Reviewer #2: Yes

Reviewer #3: Yes

2. Has the statistical analysis been performed appropriately and rigorously? 

Reviewer #1: Yes

Reviewer #2: Yes

Reviewer #3: Yes

3. Have the authors made all data underlying the findings in their manuscript fully available?

Reviewer #1: Yes

Reviewer #2: No

Reviewer #3: No

4. Is the manuscript presented in an intelligible fashion and written in standard English?

Reviewer #1: Yes

Reviewer #2: Yes

Reviewer #3: Yes

5. Review Comments to the Author

Reviewer #1: Thank you for the opportunity to review this manuscript. I believe the manuscript has the potential to make a strong contribution to the literature, and it's strengths include it's large sample and focus on gender nonbinary, agender, and asexual individuals, who are not well-represented in the literature. My specific comments for ways to improve the manuscript are included below:

1. The abstract is quite long. I would recommend decreasing the amount of background information and focusing more on the study aims and findings.

2. Page 5, line 82: I would refrain from labeling these sexual orientations as “newer,” as they have most likely existed for much longer than they were more widely known and studied, although people possessing these identities may not have used the labels that we use for them now. I think the following sentence summarizes their status more accurately by emphasizing their newness to mainstream awareness, rather than implying that they didn’t previously exist.

3. The authors mention that the data collection procedures were described in other studies. However, it seems pertinent to include at least a brief description of how these participants were recruited, since the authors emphasize the population-based nature of the sample as a strength of the study.

4. Were other demographic data collected, other than age, sexual orientation, and gender identity? Is there any information on race/ethnicity, level of education, or income? Similar to my comment above, providing this information would further bolster the authors’ argument of the representativeness of their sample.

5. Were the measures of anxiety and depression administered in English? I noticed that the participants were all native Finnish speakers. If measures were administered in Finnish, what procedures were used to validate the Finnish version of the measures?

6. Please include any anchors that were included in the Likert-type scale for the measures of anxiety and depression.

7. Since the authors note that statistical procedures were used to account for the dependence of observations from participants who were genetically related, I think it would be helpful to provide additional details about the make-up of the sample. The sample is a subset of twins and their siblings, but what proportion of the sample were sets of twins? How many were siblings of those twins, and how many were individuals who were unrelated to any other participants in the sample? How were the levels of analyses defined (e.g., was there an individual level and a “family” level that grouped siblings and twins?)?

8. On page 27, the authors mention that age had a significant effect in all analyses. It would be helpful to describe this effect here as well (i.e., was age positively or negatively associated with depression and anxiety?).

9. Page 32, line 483: it does not seem appropriate to refer to “pre- and post-transition transgender individuals,” because this over-simplifies and dichotomizes what is a complex and nuanced process for individuals who chose to utilize medical treatments for gender affirmation. These occur on a spectrum that can include hormone treatments and different surgical procedures, among other forms of gender-affirming care. I would refrain from using this phrase and instead be more specific about the population you are referring to (i.e., those who have access to gender affirming medical care and those who do not).

10. A general comment regarding the review of prior studies and interpretation of your results relative to existing literature is that it seems important to acknowledge whether other studies have been conducted in Finland or elsewhere. In the introduction, it seems important to describe any reasons why previous studies conducted outside of Finland may or may not be generalizable to a Finnish sample, and in the discussion, how the results of your study should be interpreted in the Finnish sociocultural context, relative to elsewhere.

Reviewer #2: This study is a population based estimate of sexual orientation and gender orientation, and various of the mental health issues associated with these identities. It evaluates sexual orientation, gender orientation and their intersections to establish if the minority stress hypothesis is further supported with increased mental health issues within the intersection of minority status across both gender and sexual orientation.

• On the whole, this is a very interesting, well-conducted study, that I am happy to recommend publication for.

However, I have some reservations. First, the sample is not truly a population representative sample, this is a sample of twins and siblings of twins, and there are genetic factors involved in twins as well as psychological distinctions from non-twin populations. These may have influenced the rates and levels of mental health issues. Perhaps this partially explains the elevated prevalences in lesbian, gay, and bisexuality reported, or the lower prevalences binary transgender, nonbinary, and agender found?

• Lines 454 to 464: I suggest some tempering of the argument that “Contrary to previous findings [19], our results suggest that individuals belonging to both a sexual and gender minority (double minority status) do not suffer from higher rates of anxiety or depression, in comparison to those identifying as either a sexual or gender minority (single minority status). These non-significant results may stem from the small sample size of the double minority status group. However, as Meyer [53] proposed, these groups, due to their double minority status, may develop a greater resilience …”

It is important to note that this form of double minority is only one form of intersectional minority. Others exist. For instance, being within a gender minority and a neurodevelopmental minority has been observed to be important for mental health and would constitute a double minority status. Additionally, you argue above that "Belonging to a minority within a minority could potentially contribute to increased minority stress" Wouldn't this be evidence of elevated minority status? Perhaps it is the intersectional type that is important, or the amplification within a given type, or even the interaction of these two?

• Results:

• It is unclear whether pairwise comparisons reported have controlled for multiple comparisons. Some simple modelling suggests this has not been controlled for, and so I would request this be undertaken.

• Age is reported as having had a significant effect within the models, yet no results for age are presented in relation to the models. I would request that these data are presented in the Generalized Estimating Equations output.

• Minor issues

Lines 218 – 220 & 249 – 251: It is unnecessary to report sample sizes here given these are in Table 1.

Line 314: Readers are referred to Table 4 concerning pairwise comparisons. Table 4 does not really provide pairwise comparisons, while Table 5 does.

Line 472 ”sample” is mis-spelt as “smaple”

Line 488 “if” is used when “is” was meant.

Reviewer #3: Thank you for the opportunity to review this paper examining depression and anxiety among sexual and gender minorities in Finland. This paper meaningfully extends currently literature examining mental health of LGBTQ+ people by differentiating among people with different identities within this group, and presenting detailed data on previously excluded groups (i.e., bisexual and non-binary people). This paper is very well-written and clear. Below are some comments and questions that may further strengthen this excellent manuscript.

Introduction

- Can the authors provide additional citations to the first sentence of the final paragraph beginning “Recent research has aimed to extend the scope of the minority stress theory…”, as this statement is currently not cited?

- This paragraph may additionally benefit from some discussion of why people with nonbinary gender identities may experience “more intense” structural stressors, particularly given the introduction of the minority stress theory in the previous paragraph. For example, are the authors suggesting that factors such as ‘passing’ may reduce binary transgender individuals’ experience of stigma, and therefore of depression and anxiety?

Prevalence and Definitions

- Can the authors provide a statement regarding whether sexual and gender minority prevalence statistics are available in Finland?

- Throughout this paper, people who identify as queer are included in the emerging identities (EI) category; however, in many social contexts, queer may be used as an umbrella term that may encompass other sexual minority identities (i.e., “queer and trans” as an alternative umbrella term for “sexual and gender minorities”). Can the authors clarify the use of queer as an emerging identity and whether these shifting social understandings of queer may shape how the notion of EI is operationalized?

Method

- In Table 1, the authors do not differentiate between cisgender men and cisgender women. I would be interested to hear the rationale for this, and whether the authors considered a gender-based analysis that also differentiated between cisgender individuals’ experiences of mental health challenges (i.e., gay cisgender men versus lesbian cisgender women; bisexual cisgender men versus bisexual cisgender women).

- Can the authors clarify if the survey was originally conducted in Finnish, and identify any notable features of translating sexuality and gender identities to English?

- It is interesting that this data is drawn from a survey that focused on families with twins and their siblings. Can the authors clarify what percentage of respondents were twins themselves, and potentially reflect on whether this sampling may have impacted findings?

Results

- There are no general demographic details provided for this study’s participants, which limits the reader’s interpretation of findings related to sexual and gender minority status and mental health. For example, can the authors detail the age range, income, education level, etc. of respondents?

6. PLOS authors have the option to publish the peer review history of their article (what does this mean?). If published, this will include your full peer review and any attached files.

Reviewer #1: No

Reviewer #2: **Yes: **Mark A. Stokes

Reviewer #3: **Yes: **Allie Slemon

---

## [Author Response · Author response to Decision Letter 0]

23 Aug 2022

Dear editor and dear reviewers,

Thank you for your encouraging feedback on our manuscript. We are delighted to have the opportunity to hand in a revised version, and we hope that we have managed to meet all the required revision requests and questions in an accurate and clear manner. Below you can find a list of the comments and how we have addressed them. Should we have missed anything or should any part or the manuscript still require changes, we will gladly address them and make further improvements. In addition, if anything seems to need further clarification on our part, we will do our best to answer whatever questions the manuscript may raise.

Journal Requirements:

Response: We have made adjustments in the manuscript style and formatting (see e.g., the Author note and correspondence information on the title page and the title formatting for the Supporting information captions section on the last page of the manuscript). We hope we’ve managed to detect and correct all previous mistakes. We also took the liberty of adding an Acknowledgements section to our manuscript, as we realized we had forgotten it from the previous version. We hope it is okay to still include one this late in the process.

Upon re-submitting your revised manuscript, please upload your study’s minimal underlying data set as either Supporting Information files or to a stable, public repository and include the relevant URLs, DOIs, or accession numbers within your revised cover letter. For a list of acceptable repositories, please see http://journals.plos.org/plosone/s/data-availability#loc-recommended-repositories . Any potentially identifying patient information must be fully anonymized.

Response: We suggest that the following text would be added to our data availability statement. If more details are still called for or if you have any further questions about the data and its availability, we are happy to address them. 

The data used in this study cannot be shared publicly because of the sensitivity of the research topic. The data includes information on sexual orientation, gender identity, the number of siblings each participant has, and twin status (twin/singleton). Each family is assigned its own code in the data (i.e., one that is shared between all members of the same family), it means that it could be theoretically possible to identify individual participants in some cases. Finland is a small country (with a population of ca. 5.3M people), so having access to data revealing a person’s age, sexual orientation, gender identity, and twin status in combination with knowledge about how many twin and non-twin siblings they have and what the siblings’ ages, gender identities, and sexual orientations are can already compromise the anonymity of certain participants and families.

Thus, we have opted not to share data publicly but data can be made available on request (please contact the corresponding author Marianne Källström at mkallstr@abo.fi or +358 469217716) to researchers who meet the criteria for access to confidential data as well as editorial staff at journals. For other questions and non-author affiliated data secrecy and data handling matters, please contact the Åbo Akademi University Data Protection Officer Anna-Maria Nordman (anna-maria.nordman@abo.fi; https://www.abo.fi/en/processing-of-personal-data-at-abo-akademi-university/)

Additional Editor Comments:

1. In respect to the hypotheses, I feel these could be more clearly operationalised. For example, in hypothesis 5, instead of saying “….will report more depression and anxiety” it would be more accurate to state “….will report higher symptom scores” or similar. With respect to the other hypotheses, it would be better state that you are measuring anxiety and depression symptoms, rather than ‘higher levels of depression and anxiety’.

Response: This is an excellent suggestion. We have changed the wording in the hypotheses and we agree that they sound more accurate and clear now (please see lines 277-292). We also changed the wording in Table 4 and 5, so that they also refer to symptom scores instead of depression and anxiety rates.

2. Is the survey cross-sectional or longitudinal? Please state this in the Methods. Was informed consent provided online, in person, or pen-and-paper? How were the participants recruited?

Response: A subset of the invited individuals (n = 7,716) had previously participated in similar data collections by our research group and had consented to being invited to participate in future studies. Despite some participants having taken part in earlier partly similar surveys conducted by the same researchers, the current study was cross-sectional in its design and only data collected in the above-mentioned survey was used for our analyses. All Finnish twins and siblings of twins who were over 18 years old and currently residing in Finland were identified from the Finnish Central Population Registry (http://dvv.fi/en). If their mother tongue was listed as Finnish in the Central Population Registry, they were sent an invitation letter by mail to participate in our study. A total of 33,211 invitations to participate were sent out, and 9,564 individuals (28.8%) responded by opening the survey with their participant code. The survey was filled in via an online survey platform with the help of an individual participant code that was provided in the invitation letter. The participants who agreed to their data being used for scientific purposes indicated their written and informed consent via the online survey platform prior to commencing the survey. We added these above-mentioned details to the Participants section (please see lines 305-320).

3. Out of interest, did the survey capture how long/for what duration individuals had identified as a particular gender or sexuality? One would expect higher levels of depression and/or anxiety symptoms for those who had recently chosen to identify in a particular way versus those who were more settled into their identity.

Response: We did not ask for this information, but it is quite true that having access to how long the participants’ identities have been stable could have opened up for interesting additional perspectives. The questions we used for sexual orientation were “Describe your sexual orientation” (options: straight, gay/lesbian, bisexual, other, what? [� freetext answer]). For gender identity, we only asked the questions included in the manuscript paragraph Operationalization of gender identity (please see lines 382-392). In other words, we only found out whether the participants identified with the gender they had been registered to have according to the Central Population Registry and what gender identity label they would use to describe themselves.

4. Did the survey capture any information about why participants may or may not be experiencing mental ill-health? (e.g., trauma, violence, bullying etc?). If so, I would suggest including these in your analysis via regression modelling to determine if gender and sexuality minority remained independently associated with depression and anxiety symptoms. This information would help build a fuller discussion, which is currently full of hypothesizing about the stress load of minority groups as the authors have not looked at potential explanatory factors.

Response: Sadly, we did not have information on these factors, so they could not be included in our analyses. However, we are currently working on a data collection that includes questions on childhood maltreatment and hope to be able to study this topic more in the future.

5. As per other reviewers’ comments, please be more explicit in reporting age adjustments in the results section.

Response: We have made what we hope are sufficient additions and modifications in order to make the role of age more clear in the methods and results sections, but we are happy to answer any further questions or add any more detailed information about the age adjustments if deemed appropriate (please see lines 471-476, 509-513, 534-537, 561-565, 578-581).

6. Tables 7 and 8 don’t appear to add a lot to the paper. At the very least, I would suggest removing them to a supplementary materials section and briefly reporting them in text if the authors feel this information is needed.

Response: We have moved the tables to a supporting material file (Please see the file S3 Table).

Review Comments to the Author

Reviewer #1: 

Thank you for the opportunity to review this manuscript. I believe the manuscript has the potential to make a strong contribution to the literature, and it's strengths include it's large sample and focus on gender nonbinary, agender, and asexual individuals, who are not well-represented in the literature. My specific comments for ways to improve the manuscript are included below:

1. The abstract is quite long. I would recommend decreasing the amount of background information and focusing more on the study aims and findings.

Response: Thank you for pointing this out. We have decreased the background information and shifted the focus more to the aims and results, and we feel it now captures the main message of the study in a more focused manner.

2. Page 5, line 82: I would refrain from labeling these sexual orientations as “newer,” as they have most likely existed for much longer than they were more widely known and studied, although people possessing these identities may not have used the labels that we use for them now. I think the following sentence summarizes their status more accurately by emphasizing their newness to mainstream awareness, rather than implying that they didn’t previously exist.

Response: We removed the word “newer” as suggested, and we agree that this made the text more accurate.

3. The authors mention that the data collection procedures were described in other studies. However, it seems pertinent to include at least a brief description of how these participants were recruited, since the authors emphasize the population-based nature of the sample as a strength of the study.

Response: We included some more information on the data collection procedure (please see lines 305-320). Hopefully this gives a clearer picture of the process. We are happy to add more details if needed.

4. Were other demographic data collected, other than age, sexual orientation, and gender identity? Is there any information on race/ethnicity, level of education, or income? Similar to my comment above, providing this information would further bolster the authors’ argument of the representativeness of their sample.

Response: We have not asked the participants to report on their income, level of education or race/ethnicity. We received our participants’ addresses, names, birth dates and information about their siblings from the Central Population Registry of Finland when we started our data collection (http://dvv.fi/en). The Registry does not collect or store data on race/ethnicity. In general, Finland is a very ethnically/racially homogenous country, especially when looking at the proportion of Finnish citizens who have Finnish as their native language (https://www.stat.fi/tup/suoluk/suoluk_vaesto_en.html ). As our survey was conducted in Finnish, only participants who spoke Finnish as their mother tongue were included in the sample of people we contacted. Concerning education, Finland and the rest of the Nordic countries stand out with our easily accessible and free education. This decreases the degree of inequality and segregation. The proportion of people with a post-comprehensive education is very high in international comparison, as is the proportion of people who have a tertiary (i.e., university-level) education (e.g., Statistics Finland: https://stat.fi/en/statistics/vkour ). Differences in income are also remarkably small in Finland compared to many countries, which also makes poverty rare when compared to most countries (https://stat.fi/en/statistics/tjt ). We added some text mentioning the scarceness of demographics information as a limitation (please see lines 788-792).

5. Were the measures of anxiety and depression administered in English? I noticed that the participants were all native Finnish speakers. If measures were administered in Finnish, what procedures were used to validate the Finnish version of the measures?

Response: The whole survey was administered in Finnish, which means that the measures for depression and anxiety have been translated from English. We used a back-translation procedure, where the original BSI-18 items were first translated into Finnish by an experienced psychology researcher with relevant expertise and a degree from a British university. The questions were then back-translated into English by a clinical psychologist with a PhD and 20 years of experience in mental health research. The back translation was compared with the original BSI-18 and no significant loss of information was observed. We added a mention of the translation procedure in the manuscript (please see lines 419-422).

The internal consistencies of the anxiety (α = .88) and depression (α = .87) subscales in our sample seem to match or even exceed the alpha-levels observed in other, validated translations described in peer-reviewed publications (see e.g. https://10.1023/a:1013097816238 & https://doi.org/10.1080/15305050701808680). Members of our research group have used these same measures in previous articles, which have been published in respectable peer-reviewed journals (see e.g., https://doi.org/10.1038/s41598-018-34138-8 & https://doi.org/10.1371/journal.pone.0177674). Taken together, we have interpreted these signs as sufficiently reassuring of the fact that our translations work and can be used for research purposes.

6. Please include any anchors that were included in the Likert-type scale for the measures of anxiety and depression.

Response: We added the anchors in parentheses (please see lines 415-416).

7. Since the authors note that statistical procedures were used to account for the dependence of observations from participants who were genetically related, I think it would be helpful to provide additional details about the make-up of the sample. The sample is a subset of twins and their siblings, but what proportion of the sample were sets of twins? How many were siblings of those twins, and how many were individuals who were unrelated to any other participants in the sample? How were the levels of analyses defined (e.g., was there an individual level and a “family” level that grouped siblings and twins?)?

Response: We included information about the frequencies of twins and siblings in the methods section (please see lines 324-340), as well as a description of how relatedness was coded in the analyses (please see lines 429-431).

8. On page 27, the authors mention that age had a significant effect in all analyses. It would be helpful to describe this effect here as well (i.e., was age positively or negatively associated with depression and anxiety?).

Response: We added information in the results section about this (please see lines 471-476, 509-513, 534-537, 561-565, 578-581). 

9. Page 32, line 483: it does not seem appropriate to refer to “pre- and post-transition transgender individuals,” because this over-simplifies and dichotomizes what is a complex and nuanced process for individuals who chose to utilize medical treatments for gender affirmation. These occur on a spectrum that can include hormone treatments and different surgical procedures, among other forms of gender-affirming care. I would refrain from using this phrase and instead be more specific about the population you are referring to (i.e., those who have access to gender affirming medical care and those who do not).

Response: We understand this viewpoint, thank you for bringing it up. We tried our best to rephrase this paragraph to make more accurate and nuanced in its present form (please see lines 823-831). We are open to further questions or suggestions for improvement if needed.

10. A general comment regarding the review of prior studies and interpretation of your results relative to existing literature is that it seems important to acknowledge whether other studies have been conducted in Finland or elsewhere. In the introduction, it seems important to describe any reasons why previous studies conducted outside of Finland may or may not be generalizable to a Finnish sample, and in the discussion, how the results of your study should be interpreted in the Finnish sociocultural context, relative to elsewhere.

Response: Finland and the other Nordic countries stand out internationally for a few reasons. Our health care system works well when compared to many other countries, income differences are small thanks to the well-functioning system of social welfare services and free education, and Finland has been consistently rated as a country of social progress and equality. We added some text explaining these themes, highlighting why Finnish samples could hypothetically differ from the samples in previous studies which have mostly been conducted in the U.S. and the U.K. (see lines 253-266). We also added some more phrases about previous studies conducted in Finland and how they may be different from our study (see lines 126.130, 154-155, 162-164, 198-206, 234-237). 

Reviewer #2: 

This study is a population based estimate of sexual orientation and gender orientation, and various of the mental health issues associated with these identities. It evaluates sexual orientation, gender orientation and their intersections to establish if the minority stress hypothesis is further supported with increased mental health issues within the intersection of minority status across both gender and sexual orientation.

On the whole, this is a very interesting, well-conducted study, that I am happy to recommend publication for.

However, I have some reservations. 

1. First, the sample is not truly a population representative sample, this is a sample of twins and siblings of twins, and there are genetic factors involved in twins as well as psychological distinctions from non-twin populations. These may have influenced the rates and levels of mental health issues. Perhaps this partially explains the elevated prevalences in lesbian, gay, and bisexuality reported, or the lower prevalences binary transgender, nonbinary, and agender found?

Response: While a few differences between twins and non-twins have been detected, such as birth weight and early motor and verbal development, for nearly all (psychologically) relevant phenotypes, twins are in fact comparable to non-twins (see e.g., Knopik, Niederhiser, DeFries, & Plomin (Eds.). Behavioural Genetics (7th ed.). Worth Publishers). Furthermore, the differences that have been detected (e.g., twins having lower birth weight) dissipate with age and are only rarely of such a nature that they would make it questionable to generalize findings from twins to the whole population (see also e.g., https://doi.org/10.1046/j.1466-9218.2000.00027.x ). We therefore find it tenable to propose that the present sample is in fact population-based regarding the variables we have measured and are interested in. In fact, this observation of twins being largely similar to non-twins for most psychological traits has been a core assumption underlying all twin research. To our knowledge, there are no reports of anxiety, depression, sexual or gender diversity being more prevalent among twins, but if there are any findings we have missed, we will of course be glad to address any questions or comments they may give rise to.

The reviewer raises another important point, which is whether the prevalence rates of the minority identities or the anxiety and depression group means could be affected by genetic effects between the twins. Firstly, concerning the depression and anxiety symptom scores there is no risk of relatedness affecting the results, as we chose multilevel regression analyses (i.e., Generalized Estimating Equations, GEE) to appropriately control for any confounds introduced by familial resemblance in our model. In order to check that the prevalence rates of gender and sexual minorities were not inflated by our using a sample consisting of twins and their siblings, we also ran an additional analysis where we only included one person per family (resulting in n = 6,007) and re-computed the frequencies of the different sexual and gender minority categories in a sample where none of the participants had any relation to each other. The percentages were nearly identical to the ones computed from the whole sample for heterosexual (88.1% for the subsample vs 88.7% for the whole sample), gay/lesbian (2.8% for the smaller subsample vs 2.7% for the whole sample), bisexual (7.4% for the smaller subsample vs 7.3% for the whole sample), EI (1.0% for the smaller subsample vs 1.2% for the whole sample), asexual (0.6% for the smaller subsample vs 0.7% in the whole sample), and for gender minority participants (0.9% in both the smaller subsample and the whole sample). We have not included these results in the manuscript or in the supplement because only including one person per family in our analyses would lead to lower statistical power, but we are glad to do so if it would make the manuscript more clear or transparent.

2. Lines 454 to 464: I suggest some tempering of the argument that “Contrary to previous findings [19], our results suggest that individuals belonging to both a sexual and gender minority (double minority status) do not suffer from higher rates of anxiety or depression, in comparison to those identifying as either a sexual or gender minority (single minority status). These non-significant results may stem from the small sample size of the double minority status group. However, as Meyer [53] proposed, these groups, due to their double minority status, may develop a greater resilience …”

It is important to note that this form of double minority is only one form of intersectional minority. Others exist. For instance, being within a gender minority and a neurodevelopmental minority has been observed to be important for mental health and would constitute a double minority status. Additionally, you argue above that "Belonging to a minority within a minority could potentially contribute to increased minority stress" Wouldn't this be evidence of elevated minority status? Perhaps it is the intersectional type that is important, or the amplification within a given type, or even the interaction of these two?

Response: Thank you for highlighting these very important perspectives. We included some discussion about how complex measuring or defining elevated minority status is and what the limitations of analyses therefore are (please see lines 680-682 and 752-778). We believe that the discussion is much better now concerning this topic, but we are glad to answer further questions or receive feedback on what might still be improved.

Results:

3. It is unclear whether pairwise comparisons reported have controlled for multiple comparisons. Some simple modelling suggests this has not been controlled for, and so I would request this be undertaken.

Response: Thank you for this well-motivated comment. We decided to use Bonferroni-corrected alpha values in order to correct for multiple comparisons (see lines 450-454 in the Methods section). In the Results and Discussion sections, we changed the text where needed, based on which analyses did not remain significant after controlling for multiple tests (see e.g., lines 477-486). We also included a brief reflection about correcting for multiple testing in the discussion section (please see lines 850-859). 

4. Age is reported as having had a significant effect within the models, yet no results for age are presented in relation to the models. I would request that these data are presented in the Generalized Estimating Equations output.

Response: We added information in the results section about this (please see lines 471-476, 509-513, 534-537, 561-565, 578-581). 

5. Minor issues

Lines 218 – 220 & 249 – 251: It is unnecessary to report sample sizes here given these are in Table 1.

Response: We removed the sentences where we report the sample sizes.

6. Line 314: Readers are referred to Table 4 concerning pairwise comparisons. Table 4 does not really provide pairwise comparisons, while Table 5 does.

Response: We changed the sentence to “See Table 4 and Table 5 for means and mean differences between the”, not mentioning pairwise comparisons. We also changed the titles to Tables 4 and 5, naming them “Comparisons of…”

7. Line 472 ”sample” is mis-spelt as “smaple”

Response: We corrected the spelling, thank you for noticing the mistake.

8. Line 488 “if” is used when “is” was meant.

Response: We corrected the spelling error.

Reviewer #3: 

Thank you for the opportunity to review this paper examining depression and anxiety among sexual and gender minorities in Finland. This paper meaningfully extends currently literature examining mental health of LGBTQ+ people by differentiating among people with different identities within this group, and presenting detailed data on previously excluded groups (i.e., bisexual and non-binary people). This paper is very well-written and clear. Below are some comments and questions that may further strengthen this excellent manuscript.

Introduction

1. Can the authors provide additional citations to the first sentence of the final paragraph beginning “Recent research has aimed to extend the scope of the minority stress theory…”, as this statement is currently not cited?

Response: We added more citations and briefly described the findings in a couple of sentences (please see lines 76-88).

2. This paragraph may additionally benefit from some discussion of why people with nonbinary gender identities may experience “more intense” structural stressors, particularly given the introduction of the minority stress theory in the previous paragraph. For example, are the authors suggesting that factors such as ‘passing’ may reduce binary transgender individuals’ experience of stigma, and therefore of depression and anxiety?

Response: We added some text to this part of the introduction, and we feel it is now clearer (please see lines 77-87). 

Prevalence and Definitions

3. Can the authors provide a statement regarding whether sexual and gender minority prevalence statistics are available in Finland?

Response: There are no official statistics provided by government-run agencies, and to our knowledge, there are no peer-reviewed papers on the topic. Some population-based data on adolescents in secondary school exist, but the age of the participants is quite different from the ages in the present study (please see lines 126-130, 154-155, 162-164). 

4. Throughout this paper, people who identify as queer are included in the emerging identities (EI) category; however, in many social contexts, queer may be used as an umbrella term that may encompass other sexual minority identities (i.e., “queer and trans” as an alternative umbrella term for “sexual and gender minorities”). Can the authors clarify the use of queer as an emerging identity and whether these shifting social understandings of queer may shape how the notion of EI is operationalized?

Response: We chose the term EI based on a few recent papers studying similar topics in similar populations (most notably Borgogna et al., 2019; http://dx.doi.org/10.1037/sgd0000306 ). Our choice of wording relies mostly on a practical decision of wanting to find a word that would capture the diverse sexual orientations of people who report that they do not identify with being either straight, gay/lesbian or bisexual. One term we could have used to describe this group is queer. However, as some of the people in this EI group called themselves queer and most others decided to report other terms instead (e.g., pansexual, demisexual, etc), we find that it would have been slightly inaccurate to refer to them using an umbrella term that is synonymous with what part (but not all) of this subsample described themselves as. Consequently, we felt it would be conceptually clearer to use an umbrella term that was not interchangeable with one of the subgroup labels in this larger group of diverse sexual minorities. Furthermore, even though queer is sometimes used to describe gender identity (or both one’s sexual orientation and gender identity), the participants who described themselves as queer in the present study chose to do so in response to the open-ended question “How would you describe you sexual orientation?”. Additionally, none of the respondents of the present study used the term “queer” to describe their gender identity when offered the possibility to do so in a similar open-ended question item where they could type in the exact words they use to define their gender. The usage of the term queer surely varies based on social context, so it is of course important to bear in mind that the term queer might be used slightly differently in different countries and contexts. We added some text that hopefully clarifies why we chose the term EI (please see lines 358-371), but we are open to any further comments or suggestions on how to further improve the manuscript on this point.

Method

5. In Table 1, the authors do not differentiate between cisgender men and cisgender women. I would be interested to hear the rationale for this, and whether the authors considered a gender-based analysis that also differentiated between cisgender individuals’ experiences of mental health challenges (i.e., gay cisgender men versus lesbian cisgender women; bisexual cisgender men versus bisexual cisgender women).

Response: Dividing some groups but not others would have led to us comparing single sex groups (e.g., bisexual women) and mixed groups (EI individuals of all genders) to each other within the same analyses. We therefore chose to not differentiate any of the groups for the sake of coherence. We added some text to the manuscript mentioning this as a limitation (please see lines 793-799).We also added information of the proportion of female and male participants in the manuscript so that the reader could still access information about the proportion of female and male participants in the sample (please see lines 384-386). We are glad to address any further questions on this choice, if our answer raises any comments

6. Can the authors clarify if the survey was originally conducted in Finnish, and identify any notable features of translating sexuality and gender identities to English?

Response: Thank you for this comment, we agree that this part is not evident and would benefit from clarification. The survey was conducted in Finnish, and the translation of terminology from Finnish to English was made by the authors. We did not encounter any problems or challenges with the translation part, as many of the Finnish terms are direct and literal translations from English (e.g., “sukupuoleton” literally means “without gender”, and is clearly seen as translatable to the term agender in English). Some other terms are even direct English words, so no translation was needed (e.g. in the context of sexual orientation and gender diversity, “queer” is used in the same way in the Finnish language as in English). We added a sentence where we specify the language of the survey and describe the translation process (please see the first paragraph in the S1_File supplement).

7. It is interesting that this data is drawn from a survey that focused on families with twins and their siblings. Can the authors clarify what percentage of respondents were twins themselves, and potentially reflect on whether this sampling may have impacted findings?

Response: We included more information about the sample and the proportion of twins and non-twin siblings (please see lines 324-340). Using twin samples to study psychological constructs has been widely and critically discussed. To our knowledge however, using twins to study most psychological constructs is a valid choice and results should not be impacted by the respondents being twins (https://doi.org/10.1046/j.1466-9218.2000.00027.x ). The only cases in which differences between twins and singletons may emerge and impact findings is when one studies young children, as potential differences in birth weight may affect language and motor development in the early years of development. Also, as many in the sample were genetically related to each other, this relatedness has to be controlled for in order for it to not affect the results of our analyses (please see the Analyses section of the manuscript and the answer to question 1 by reviewer 2)

Results

8. There are no general demographic details provided for this study’s participants, which limits the reader’s interpretation of findings related to sexual and gender minority status and mental health. For example, can the authors detail the age range, income, education level, etc. of respondents?

Response: Thank you for bringing this up. Sadly, the demographic information available on our participants is quite scarce. We know the exact ages of our participants, as this was part of the information we gained from the Finnish Central Population Registry prior to contacting prospective participants. Apart from the age and home address of our participants, no other demographic information was collected. We have included the age range of the participants in the Methods section (please see lines 323-324). We also added some text mentioning the lack of large-scale demographics information as a limitation (please see lines 788-792). Please see the response to question 4 posed by Reviewer 1 for some additional details on the general demographics of the Finnish population.

---

## [Decision Letter · Decision Letter 1]

20 Sep 2022

PONE-D-22-08353R1Mental health among sexual and gender minorities: A Finnish population-based study of anxiety and depression discrepancies between individuals of diverse sexual orientations and gender minorities and the majority populationPLOS ONE

Dear Dr. Källström,

Thank you for submitting your manuscript to PLOS ONE. After careful consideration, we feel that it has merit but does not fully meet PLOS ONE’s publication criteria as it currently stands. Therefore, we invite you to submit a revised version of the manuscript that addresses the points raised during the review process.

I wish to thank the authors for their considered responses to the original reviewer comments. Some additional minor issues have been noted in the revision process, which are aligned to the readability of the manuscript.  Please ensure that your decision is justified on PLOS ONE’s publication criteria and not, for example, on novelty or perceived impact.

We look forward to receiving your revised manuscript.

Kind regards,

Michelle Torok, Ph.D.

Academic Editor

PLOS ONE

Journal Requirements:

Reviewers' comments:

Reviewer's Responses to Questions

**Comments to the Author**

1. If the authors have adequately addressed your comments raised in a previous round of review and you feel that this manuscript is now acceptable for publication, you may indicate that here to bypass the “Comments to the Author” section, enter your conflict of interest statement in the “Confidential to Editor” section, and submit your "Accept" recommendation.

Reviewer #1: All comments have been addressed

Reviewer #2: All comments have been addressed

2. Is the manuscript technically sound, and do the data support the conclusions?

Reviewer #1: Yes

Reviewer #2: Yes

3. Has the statistical analysis been performed appropriately and rigorously? 

Reviewer #1: Yes

Reviewer #2: Yes

4. Have the authors made all data underlying the findings in their manuscript fully available?

Reviewer #1: No

Reviewer #2: Yes

5. Is the manuscript presented in an intelligible fashion and written in standard English?

Reviewer #1: Yes

Reviewer #2: Yes

6. Review Comments to the Author

Reviewer #1: Thank you for your work to address my previous comments. I believe the manuscript has been improved as a result of the authors' responses to mine and other reviewers' comments. 

As a side note, it looks like something happened with formatting of the manuscript such that the line numbers the authors gave for revisions were inaccurate. In any future revisions, it would be helpful to ensure that the line numbers in the responses to reviewer comments are consistent with where changes are located in the manuscript.

Here are the few additional comments that authors should address:

1. The paragraph on page 11 is very long. Please divide into multiple paragraphs.

2. The paragraph on pages 16-17 is also very long. Please divide into multiple paragraphs.

3. The authors noted in a response to another comment in the original revision that the population of Finland is homogenous in regard to race/ethnicity, education, and income level. I think this information would be important to include in the discussion of how the results of this study should be interpreted, particularly given that the authors discuss the broader implications of intersectionality on page 32. The homogeneity of the Finnish population would seem to indicate that the results are most generalizable to a similar population.

Reviewer #2: I thank the authors for their attention to my concerns. No further changes are sought by me in respect of this manuscript.

7. PLOS authors have the option to publish the peer review history of their article (what does this mean?). If published, this will include your full peer review and any attached files.

Reviewer #1: No

Reviewer #2: No

---

## [Author Response · Author response to Decision Letter 1]

29 Sep 2022

Dear editor and dear reviewers,

Thank you once again for your feedback on our manuscript. We are happy to hand in this revised version that we feel corresponds to the changes asked for in the previous round of reviews. Below you can find a list of the comments we have received and how we have addressed them. Should we have missed anything or should any part or the manuscript still require changes, we will gladly address them and make further improvements. In addition, if anything seems to need further clarification on our part, we will do our best to answer whatever questions the manuscript may raise.

Journal Requirements:

Response: Thank you for bringing this up. We went through the references section with special attention, and we did find a number of small formatting and spelling errors which we have now corrected. The changes are listed below. We believe that the reference list is now complete and correct, but if there is anything in it that seems to still require revisions we will of course go through the list again and make the required changes. 

Reference 1: We changed the format of the reference to match the guidelines for PlosONE.

Reference 16: We deleted an unnecessary period from the end of the doi-number.

Reference 20: We changed the format of the reference to match the guidelines for PlosONE.

Reference 24: We realized that the doi-number of the reference and the name of the first author were correct, but not the other details. This means that we intended to refer to a another, slightly differently themed article written by M. Paz Galupo (published in another journal). We corrected the reference information and left the first author and the doi-number as they were. We sincerely apologize for this mistake.

Reference 28: We changed the formatting of the reference to match the guidelines for PlosONE (according to the instructions on how to cite a monograph on the internet).

Reference 29: We added a missing page number and a period to the reference.

Reference 35: We corrected the capitalization of the word “national” (National –> national)

Reference 37: We corrected the abbreviation of the journal name.

Reference 43: We deleted an unnecessary period from the end of the doi-number and we corrected the formatting of the author names.

Reference 48: We added a missing page number.

Reference 50: We corrected the spelling of the last author’s name.

Reference 53: We corrected two errors in punctuation.

Reference 56: We corrected a spelling error and added a missing comma.

Reference 61: We corrected the formatting so that it corresponds to the instructions for referring to a monograph on the internet.

Reference 62: We corrected the formatting of the reference.

Reference 63: We corrected the formatting of the reference and corrected the authors and the doi-number.

Reference 67: We corrected the formatting regarding punctuation and capitalization of the study title.

Reference 69: We corrected the formatting of the reference so that it corresponds to the instructions on how to refer to a monograph on the internet.

References 70-72 were added because of the changes we made based on Reviewer 1’s comments (please see comment 3 below). This means that the manuscript’s reference section now features 74 references in total (the total number of references in the previous version was 71).

Review Comments to the Author

Reviewer #1: Thank you for your work to address my previous comments. I believe the manuscript has been improved as a result of the authors' responses to mine and other reviewers' comments. 

As a side note, it looks like something happened with formatting of the manuscript such that the line numbers the authors gave for revisions were inaccurate. In any future revisions, it would be helpful to ensure that the line numbers in the responses to reviewer comments are consistent with where changes are located in the manuscript.

Response: Thank you, we are delighted to hear that the changes we have made based on the previous comments seem to have improved the quality of the manuscript. We would also like to apologize for the mix-up with the line numbers. We will pay special attention to this in the future.

Here are the few additional comments that authors should address:

1. The paragraph on page 11 is very long. Please divide into multiple paragraphs.

Response: We divided the original paragraph in three, and we also found one error in punctuation that we corrected (“A subset of the invited individuals (n = 7,716) had previously participated in similar data collections by the same research group and had consented to being invited to participate in future studies [56], Despite some”). The comma after the word “studies” was changed into a period (“.”). Please see page 11 in the manuscript.

2. The paragraph on pages 16-17 is also very long. Please divide into multiple paragraphs.

Response: We divided the original paragraph into three shorter paragraphs (please see p 16-17).

3. The authors noted in a response to another comment in the original revision that the population of Finland is homogenous in regard to race/ethnicity, education, and income level. I think this information would be important to include in the discussion of how the results of this study should be interpreted, particularly given that the authors discuss the broader implications of intersectionality on page 32. The homogeneity of the Finnish population would seem to indicate that the results are most generalizable to a similar population.

Response: We agree, it would have been good to add these points into the manuscript itself as well. We added these points of discussion to the beginning of the Strengths and Limitations section (please see p 33).

Reviewer #2: I thank the authors for their attention to my concerns. No further changes are sought by me in respect of this manuscript.

Response: Thank you for your feedback, we really appreciate your work with reviewing this manuscript and helping us improve it.

Thank you once again for your valuable comments and for the opportunity to have our work reviewed at Plos One. 

Sincerely, on behalf of my co-authors,

Marianne Källström

---

## [Editor Report · Decision Letter 2]

10 Oct 2022

Mental health among sexual and gender minorities: A Finnish population-based study of anxiety and depression discrepancies between individuals of diverse sexual orientations and gender minorities and the majority population

PONE-D-22-08353R2

Dear Dr. Källström,

We’re pleased to inform you that your manuscript has been judged scientifically suitable for publication and will be formally accepted for publication once it meets all outstanding technical requirements.

Kind regards,

Michelle Torok, Ph.D.

Academic Editor

PLOS ONE